# Carbon Dioxide Gas Sensor Based on Terahertz Metasurface with Asymmetric Cross-Shaped Holes Empowered by Quasi-Bound States in the Continuum

**DOI:** 10.3390/s25134178

**Published:** 2025-07-04

**Authors:** Kai He, Tian Ma

**Affiliations:** College of Safety Science and Engineering, Xi’an University of Science and Technology, Xi’an 710054, China; 22403050214@stu.xust.edu.cn

**Keywords:** bound states in the continuum, planar metasurface, terahertz, gas sensing

## Abstract

In this paper, a novel type of polarization-insensitive terahertz metal metasurface with cross-shaped holes is presented, which is designed based on the theory of bound states in continuous media. The fundamental unit of the metasurface comprises a metal tungsten sheet with a cross-shaped hole structure. A thorough analysis of the optical properties and the quasi-BIC response is conducted using the finite element method. Utilizing the symmetry-breaking theory, the symmetry of the metal metasurface is broken, allowing the excitation of double quasi-BIC resonance modes with a high quality factor and high sensitivity to be achieved. Analysis of the multipole power distribution diagram and the spatial distribution of the electric field at the two quasi-BIC resonances verifies that the two quasi-BIC resonances of the metasurface are excited by electric dipoles and electric quadrupoles, respectively. Further simulation analysis demonstrates that the refractive index sensitivities of the two quasi-BIC modes of the metasurface reach 404.5 GHz/RIU and 578.6 GHz/RIU, respectively. Finally, the functional material PHMB is introduced into the metasurface to achieve highly sensitive sensing and detection of CO_2_ gas concentrations. The proposed metallic metasurface structure exhibits significant advantages, including high sensitivity, ease of preparation, and a high Q-value, which renders it highly promising for a broad range of applications in the domains of terahertz biosensing and highly sensitive gas sensing.

## 1. Introduction

The concept of bound states in the continuum (BIC) was initially introduced into the domain of quantum mechanics by John von Neumann and Eugene Wigner in 1929 [1]. BIC represents a unique wave phenomenon in which a resonance mode is embedded within the radiation continuum spectrum yet remains completely localized without energy leakage, leading to a theoretically infinite quality factor (Q-factor) [2,3,4]. This phenomenon breaks the traditional concept that resonant states in the continuous spectrum must radiate energy. The formation of BIC relies on symmetry protection mechanisms and coupled-mode theory [5,6,7]. The potential of BIC has been shown in a variety of fields, including optics, acoustics, and electromagnetics [8,9,10]. It can be used to achieve highly sensitive sensors [11,12] and enhance nonlinear effects [13,14]. For instance, in 2023, B. Liu et al. proposed a QBIC metasurface with a high refractive index sensitivity of 420 GHz/RIU, and its direct limit of detection (LoD) for trace homocysteine (Hcy) molecules could reach 12.5 pmol/μL [15].

In recent years, research into electromagnetic metasurfaces has focused on achieving high quality factor (high-Q) resonances [16,17]. The strong interaction between light and matter is one of the key factors in achieving this goal. Introducing the concept of BIC into metasurface systems provides an effective way to significantly enhance the interaction between light and matter [18,19]. Metasurfaces can be defined as two-dimensional arrays of sub-wavelength-scale artificial structures arranged in specific periodic configurations. Through the design of these periodic structures, targeted electromagnetic wave responses can be engineered [20,21]. However, the ideal BIC modes cannot be directly observed via spectral measurements due to the absence of radiative energy channels and their unlimited lifetimes [22]. However, through the introduction of structural perturbations to break the system’s symmetry, the BIC state couples with extended states, resulting in energy leakage and the formation of quasi-BIC states [23]. These quasi-BIC states can be observed due to their finite Q-factors and radiative losses. When structural perturbations are introduced into a metasurface structure to break its in-plane symmetry, thus generating a quasi-BIC state, such a metasurface is called a symmetry-protected BIC metasurface. However, traditional metasurface fabrication processes remain highly complex for several reasons. First, the development of sub-wavelength-scale structural features necessitates the employment of advanced micro/nanofabrication techniques, such as electron-beam lithography or photolithography, which impose stringent requirements on the precision of equipment and operational conditions. Second, the fabrication process involves many steps, including substrate pretreatment, deposition of base and metallic layers, pattern replication and transfer, etching, and post-processing [24,25,26]. During the etching process, material- and structure-specific methods (e.g., plasma etching or reactive ion etching) must be selected, ensuring strict control over the depth and uniformity of etching. Metasurfaces based on BICs can be applied in gas sensing, such as monitoring the concentration of carbon dioxide, due to their high-Q value characteristics.

The monitoring of carbon dioxide (CO_2_)—a critical component of environmental protection, human health, industrial safety, and global sustainable development—is paramount [27]. The provision of real-time data facilitates the establishment and evaluation of carbon neutrality targets, ensures the quality of indoor air to mitigate health risks associated with hypoxia in enclosed spaces, and enables the dissemination of safety risk warnings in high-risk industrial operations while optimizing energy efficiency. As a primary greenhouse gas, elevated CO_2_ concentrations are a key driver of global temperature increases, inducing environmental challenges such as glacial retreat, sea-level rise, and ecosystem disruption. Recent measurements have indicated that the global average CO_2_ concentration has surpassed 420 ppm, exceeding pre-industrial levels by over 50% and far surpassing the safe threshold for mitigating environmental degradation [28,29]. This underscores the pressing need for highly sensitive and portable CO_2_ sensors. The prevailing gas sensing mechanisms are predominantly reliant on two approaches: evanescent field absorption [30,31,32] and resonant frequency shift detection [33,34].

This paper innovatively proposes a tungsten metal metasurface based on the arc-corner cross-hole structure and excites a polarization-independent double quasi-BIC resonant mode through symmetry breaking. The structural design adopts mechanically compatible arc-shaped cross holes instead of right-angle structures, significantly simplifying the preparation process. Theoretical simulations demonstrate that breaking the in-plane symmetry of the metasurface system enables the excitation of dual quasi-BIC resonances with ultra-high quality factors ($Q>10^{3}$) at 0.5 THz and 0.67 THz. Subsequent analysis of the electric field response under varying polarization angles and multipole decomposition at these resonant frequencies reveals that the dominant modes correspond to electric dipole and quadrupole interactions, confirming the polarization-insensitive nature of the metasurface. In order to explore sensing applications, a functional polymer layer—comprising polyhexamethylene biguanide (PHMB)—is further integrated into the metasurface for the detection of carbon dioxide (CO_2_). High refractive index sensitivity (578.6 GHz/RIU) and CO_2_ gas detection sensitivity (0.06 GHz/ppm) were achieved. Combined with the PHMB functional layer, a detection limit of 33.33 ppm was realized. This work synchronously achieves polarization-independent and dual high-Q resonance in metal metasurfaces, providing a new idea for terahertz gas sensing.

## 2. Structural Design and Simulation

Figure 1 illustrates the proposed metasurface design, with the inset in the upper right corner depicting the unit cell structure. The unit cell consists of a square lattice patterned on a tungsten metal plate with a period length of $P_{x}\;{=P}_{y}=400\;\mu m$. In the simulation software, metallic tungsten was set as lossy metal with the conductivity of $1.89\times 10^{7}S/m$. A cruciform aperture of length L=280 μm is formed by four sector-shaped notches with diagonal radii $R_{1}$ and $R_{2\;}$. The metasurface asymmetry is defined as $\delta (\delta =R_{2\;}-R_{1})$, which is determined by adjusting R_2_. In practice, δ must be selected by taking into account the resolution of the terahertz spectrometer (e.g., a frequency resolution of approximately 2 GHz) and the relevant characteristics of the analyte, ensuring that the resonance linewidth (γ) exceeds the instrument resolution. During structural optimization, $R_{1}$ was fixed at 80 μm, while $R_{2\;}$ was varied. When δ ≠ 0 μm, the mirror symmetry of the metasurface is disrupted, thereby exciting the quasi-bound state in the continuum (quasi-BIC) response. In this study, the optical properties of the proposed metasurface were analyzed using numerical simulations performed with CST Microwave Studio 2021. In these simulations, periodic boundary conditions were applied in the x- and y-directions, while the perfectly matched layer conditions were set in the z-direction. Unit cell of the proposed metasurface was immersed in a homogeneous medium with n=1 and normally illuminated by linearly polarized terahertz waves with electric components parallel to the y direction.

The choice of tungsten as the material is for ultra-surface applications. The melting point of tungsten is as high as 3422 °C, which is much higher than that of silver (962 °C) or copper (1085 °C), and the coefficient of expansion of tungsten is $4.5\times 10^{-6}$/K, which is much lower than that of copper ($6.5\times 10^{-6}/K$) and silver ($18.5\times 10^{-6}/K$), which means that tungsten has a much smaller deformation at high temperature as compared to copper and silver under the same temperature, which helps to improve the yield in the preparation process using laser cutting (DLC). Tungsten’s Mohs hardness of 7.5–8.5 is much higher than that of copper (3.0) and silver (2.5–3.0), which means that tungsten is able to support thinner super surfaces. The proposed metallic metasurface is expected to allow for the removal of its central aperture using laser cutting technology.

Different thicknesses of the metasurface change the electromagnetic field localization strength, resonance frequency, and coupling effects, resulting in significant differences. In general thinner thicknesses (<λ/10, λ is the operating wavelength) are more likely to excite surface equipartitioned exciton resonances for narrowband high Q devices, thicker structures (>λ/5) are more likely to excite Fabry–Perot (FP) cavity resonances or waveguide modes for broadband responses, and intermediate thicknesses are more likely to produce mixed resonances.

To extract the resonance frequency and linewidth of the quasi-bound state in the continuum (quasi-BIC) from the frequency-domain spectrum, the simulated results were fitted to the standard Fano resonance mode described in [35]:(1)$$T=\left|a+ib+c/(\omega -\omega_{0}+i\gamma )\right|$$

Here, a, b, and c are fitting constants associated with the Fano resonance frequency ($\omega_{0}$) and spectral linewidth (γ), where γ encompasses both radiative and ohmic losses. The quality factor (Q) of the Fano resonance under varying asymmetry parameters δ is calculated by substituting the extracted resonance frequency ($\omega_{0}$) and linewidth (γ) from Equation (1) into Equation (2). The Q-factor of the quasi-bound state in the continuum (quasi-BIC) metasurface scales inversely with the square of the asymmetry parameter δ; that is, $Q\;\propto \;\delta^{-2}$.(2)$$Q={\;\omega }_{0}/2\gamma$$

## 3. Analysis and Discussion

This study conducted a systematic investigation of the frequency-domain transmission spectra and resonance responses of a tungsten-based metasurface under varying values of the structural asymmetry parameter δ. Figure 2a presents the pseudo-colour map of transmission spectra for different δ values. At δ=0 μm, the metasurface retains C_2_ symmetry, supporting a symmetry-protected bound state in the continuum (BIC) with zero radiative leakage and theoretically infinite quality factor (Q→ ∞). As evidenced in Figure 2b, the corresponding transmission spectrum exhibits a non-radiative BIC state characterized by vanishing spectral linewidth.

Simulated transmission spectra under asymmetric conditions (δ ≠ 0 μm) are shown in Figure 3. Breaking the C_2_ symmetry enables coupling between the BIC mode and the radiative continuum, thereby generating quasi-BIC states that manifest Fano-type resonances with finite Q-factors. Specifically, dual quasi-BIC resonances were excited at 0.50 THz (Mode I) and 0.67 THz (Mode II), as demonstrated in Figure 2b. It is evident that increasing δ induces a shift in resonance frequency towards lower frequencies, accompanied by a progressive broadening of resonance linewidths. This tunability enables effective control of the metasurface’s Q-factor through modulation of δ.

As shown in Figure 3, the quality factor (Q) of the tungsten-based metasurface was determined as a function of the asymmetry parameter δ, utilizing Equations (1) and (2). At δ=20 μm, the Q-factors of the two resonant modes (Mode I and Mode II) were 145 and 42, respectively. With increasing δ, the Q-factor decreased sharply. Conversely, reducing δ allowed the Q-factor to reach values exceeding 103. This behavior primarily arises due to the inherent ohmic loss of tungsten in the terahertz regime. In order to ascertain the ideal Q-factor in the absence of material loss, we replaced tungsten with a perfect electric conductor (PEC) in simulations. The results revealed that decreasing δ significantly enhanced the Q-factor of the PEC metasurface, while increasing δ led to rapid degradation. Compared to the lossy metallic metasurface, the PEC metasurface (neglecting ohmic dissipation) exhibited substantially higher Q-factors, following an inverse-square dependence on δ (i.e., $Q\;\propto \;\delta^{-2}$*)*.

Figure 4 illustrates the transmission spectral characteristics of the metallic metasurface at a thickness of δ = 20 μm under various incident angles. As can be clearly observed from the figure, the resonance peak near 0.500 THz gradually shifts towards higher frequencies with increasing incident angle, while its resonance linewidth remains essentially constant. In contrast, for the two resonance peaks near 0.700 THz, the spacing between them progressively narrows as the incident angle increases, and their resonance linewidths exhibit a tendency to gradually decrease. When the incident angle reaches 14°, the resonance linewidths of these two peaks attain their minimum values, and the corresponding Q-factors reach their maximum values. However, as the incident angle continues to increase beyond 14°, the resonance linewidths gradually increase once again.

Figure 5 presents the transmission spectral variation patterns of the metallic metasurface with different values of structural parameters, namely the side length L of the central rectangular aperture, the period P, the thickness t, and the fillet radius R_1_. Notably, the effects of increasing the structural parameters L and P on the transmission spectrum exhibit a certain degree of similarity. As the parameter L increases, the resonance peak near 0.525 THz shifts towards lower frequencies. Similarly, with the increase in parameter P, the resonance peak near 0.550 THz also gradually moves towards lower frequencies. For the resonance peak near 0.7 THz, it slowly shifts towards lower frequencies as parameter L increases, while the resonance peak near 0.75 THz rapidly moves towards lower frequencies as parameter P increases. In contrast, the influence of the parameters t and R_1_ on the transmission spectrum is relatively minor. As the parameters t and R_1_ increase, the resonance peak near 0.500 THz slowly shifts towards higher frequencies. The resonance peak near 0.700 THz remains almost unchanged with the increase in parameter t, but it slowly moves towards lower frequencies as parameter R_1_ increases.

Figure 6 illustrates the spatial distribution of the normalized induced electric field for the metasurface in both the bound state in the continuum (BIC, δ=0 μm) and quasi-BIC (δ=20 μm) states at their respective resonant frequencies. The directions of the arrows denote the instantaneous electric field orientation. In the BIC state, Mode I and Mode II exhibit vertically aligned electric field lines within the cross-shaped apertures, propagating from bottom to top. This configuration mimics the characteristic dipole-like field distribution of electric dipole resonance. When symmetry is disrupted (δ=20 μm), the metasurface transitions to the quasi-BIC state. For Mode I, the electric field intensity increases markedly compared to the BIC state, retaining a dipole-oriented profile but reorienting horizontally (left to right). Simultaneously, Mode II demonstrates enhanced field localization with a transverse alignment (vertical to horizontal), consistent with the spatial symmetry of electric quadrupole resonance. As shown in Figure 6, the electric field distribution of the BIC state exhibits high symmetry. This symmetry prevents the modal field in the BIC state from coupling with the extended-state modal field, resulting in the absence of a resonant signal in the spectrum for the BIC state. However, when the structural symmetry is broken in an asymmetric configuration, coupling occurs between the BIC state and the extended states, leading to energy leakage along the *z*-axis. This mode leakage pathway causes the system to transition from an ideal BIC state to a quasi-BIC state, resulting in a high-Q, high-sensitivity resonant peak in the spectrum.

To elucidate the underlying resonance mechanism, multipole decomposition was performed via Cartesian expansion of the surface current density [36]. As shown in Figure 7, the contributions of electric dipole (P), magnetic dipole (M), toroidal dipole (T), electric quadrupole ($Q_{e}$), and magnetic quadrupole ($Q_{m}$) modes were quantified for both BIC (δ=0 μm) and quasi-BIC (δ=20 μm) states. In the BIC state, all multipole moments exhibited weak intensities, dominated by electric dipole contributions. When transitioning to the quasi-BIC state, Mode I at 0.500 THz presented a 10-fold enhancement in electric dipole strength, while higher-order multipoles (e.g., octupoles) remained negligible. For Mode II at 0.670 THz, the electric dipole contribution decreased by two orders of magnitude, but the electric quadrupole response increased slightly. This distinct behavior confirms that Mode I is primarily driven by electric dipole interactions, whereas Mode II arises from electric quadrupole coupling—a direct consequence of symmetry-breaking-induced mode hybridization. The dominance of electric dipole and quadrupole excitations further validates that quasi-BIC resonances in this metasurface are predominantly mediated by the electric field component of terahertz waves.

To validate the potential of the proposed tungsten-based metasurface for terahertz (THz) sensing applications, we investigated its performance under varying ambient refractive indices. Table 1 compares the sensor’s key metrics (sensitivity, figure of merit) with those of state-of-the-art designs, demonstrating its competitive advantages. Figure 8a displays the THz transmission spectra as the surrounding refractive index (n) increases from 1.0 to 1.6. Notably, minor refractive index variations (Δn=0.1) induce significant resonance frequency redshifts. A linear redshift trend can be observed for n=1.0−1.2 (Figure 8b), confirming its robust linearity within this range. Linear regression yields refractive index sensitivities of 404.5 GHz/RIU (Mode I) and 578.6 GHz/RIU (Mode II) across n=1.0−1.6. The sensor’s performance is further quantified by the figure of merit (FoM=S/∆ω, where S denotes the sensitivity, and Δω is the resonance linewidth). FoM values of 33 and 32 were calculated for Mode I and Mode II, respectively, surpassing those of conventional THz sensors (Table 1). Due to the high sensitivity and FoM of the proposed metal metasurface, it shows significant application potential; for example, as a sensing device for identifying various gas analytes and aerosols or for measuring the concentration of dust particles in the air.

Figure 9a illustrates the CO_2_ gas sensing mechanism of the metasurface functionalized with a polyhexamethylene biguanide (PHMB) polymer layer. PHMB—a basic amide-group-rich polymer—serves as a selective host material for CO_2_ adsorption [41], as its negligible response to hydrogen, nitrogen, and other interferents ensures high selectivity for CO_2_ detection [42]. Figure 9b demonstrates the linear correlation between the PHMB layer’s refractive index (n) and CO_2_ concentration [43]. This relationship is attributed to the redistribution of electron density within the PHMB layer induced by CO_2_, which alters its polarizability and consequently modulates n. As a result, the resonance frequency of the metasurface shifts proportionally to the CO_2_ concentration, enabling real-time quantitative detection. The PHMB layer thickness ($H_{PHMB}$) critically influences the sensor’s performance. As shown in Figure 10a, increasing $H_{PHMB}$ from 0 to 350μm induces a gradual redshift in the resonance frequency, with the unit frequency shift (∆f) of Mode I and Mode II decreasing exponentially. Beyond $H_{PHMB}=350\;\mu m$ (see Figure 10b), ∆f exhibits independence on $H_{PHMB}$, establishing an optimal operational thickness. Using H_PHMB_ = 350 μm, we further analysed CO_2_ concentration-dependent spectral shifts. Figure 11a reveals a blueshift in resonance frequencies for both modes as CO_2_ concentration increases, attributed to the PHMB layer’s decreasing n upon gas absorption. The quantification of this trend is demonstrated in Figure 11b, which shows a positive linear correlation (i.e., $∆f\;\propto {CO}_{2}$). The recorded sensitivities were 0.06 GHz/ppm (Mode I) and 0.04 GHz/ppm (Mode II). Sensitivity is an important performance indicator for measuring sensors. The resonant shift within a unit concentration in the terahertz band can be expressed by Equation (3):(3)$$S=\frac{∆f(GHz)}{∆conc.(ppm)}$$

The resonance frequency shift (∆f*,* in GHz) corresponds to the change in CO_2_ gas concentration ([CO_2_], in ppm). The sensitivities of Modes I and II of the metasurface were determined as 0.06 GHz/ppm and 0.04 GHz/ppm, respectively. Given the typical spectral resolution of commercially available terahertz time-domain spectrometers (2 GHz), the metasurface was found to achieve a minimum detection limit (MDL) of 33.33 ppm for CO_2_. Furthermore, the sensor exhibited a linear response to CO_2_ concentrations ranging from 0 to 1000 ppm. This operational range encompasses atmospheric CO_2_ levels, validating the applicability of the proposed metasurface for real-time environmental monitoring.

## 4. Conclusions

This study proposed a terahertz (THz) metasurface leveraging quasi-bound states in the continuum (quasi-BIC) resonances. The metasurface comprises cross-apertured tungsten unit cells arranged in a square lattice. Theoretical simulations revealed dual ultra-high quality factor ($Q>10^{6}\;and\;10^{5}$) resonances at 0.5 THz and 0.67 THz, corresponding to electric dipole and electric quadrupole modes, respectively. Multipole decomposition analysis of the electric field distribution confirmed that these quasi-BIC resonances originate from hybridized interactions between electric dipole and quadrupole moments. The multipole power distribution in the quasi-BIC state demonstrated a 10-fold enhancement compared to the BIC state. As a refractive index sensor, the metasurface achieved sensitivities of 404.5 GHz/RIU (Mode I) and 578.6 GHz/RIU (Mode II), with figures of merit (FoM) of 33 and 32, respectively. For gas sensing applications, integration of a polyhexamethylene biguanide (PHMB) functional layer was shown to enable the detection of CO_2_. When the PHMB thickness ($H_{PHMB}$) exceeds 350 μm, the metasurface’s frequency shift (Δf) exhibits a linear dependence on $H_{PHMB}$ ($R_{2}$ > 0.98). This configuration achieved a CO_2_ detection sensitivity of 0.06 GHz/ppm and a minimum detection limit (MDL) of 33.33 ppm, covering atmospheric CO_2_ concentration ranges (0−1000 ppm). These findings advance the design of high-performance THz metasurfaces for environmental sensing, nonlinear optics, and integrated photonic systems.

## Figures and Tables

**Figure 1 sensors-25-04178-f001:**
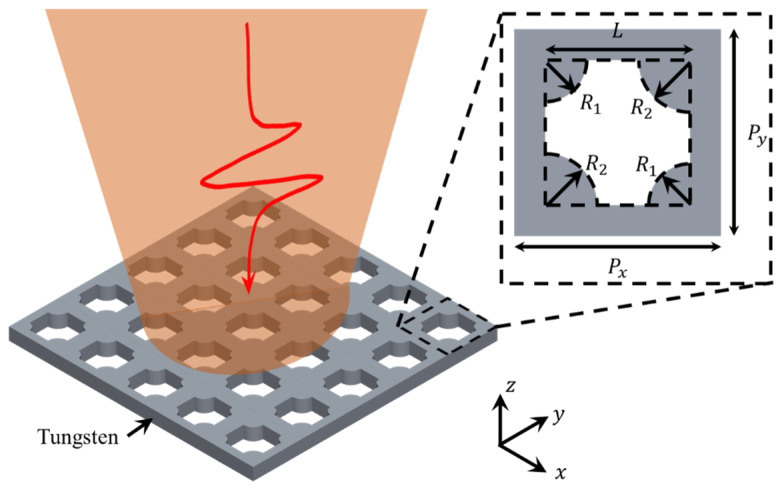
Schematic diagram of the structure of the designed metal metamaterial. The red arrow refers to a vertically incident terahertz wave, and the orange indicator refers to the covered area. Illustration: top view of the unit structure.

**Figure 2 sensors-25-04178-f002:**
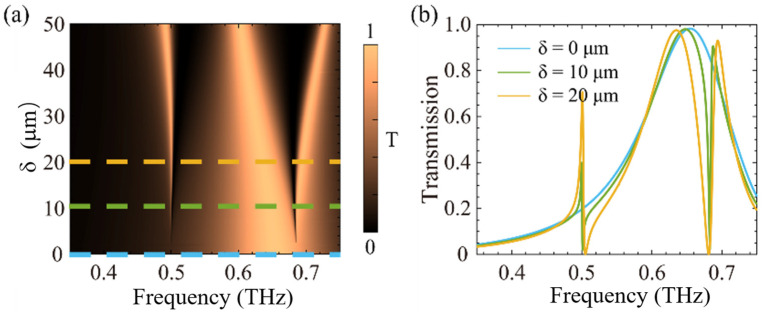
(**a**) The pseudo-colour map of transmission spectra for different δ values. (**b**) Transmission lines of metallic tungsten metamaterials for different δ values. The blue dashed line refers to the transmission spectral line at δ=0 μm, while the green dashed line corresponds to δ=10 μm, and the yellow dashed line represents δ=20 μm.

**Figure 3 sensors-25-04178-f003:**
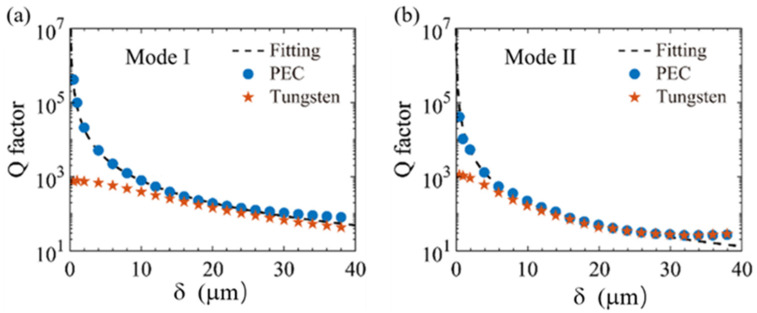
***Q factors of (a) Mode I and (b) Mode II*** with respect to the asymmetry parameter δ. The blue dots refer to the perfect electric conductor (PEC), while the orange pentagrams are the tungsten-based metallic metamaterial.

**Figure 4 sensors-25-04178-f004:**
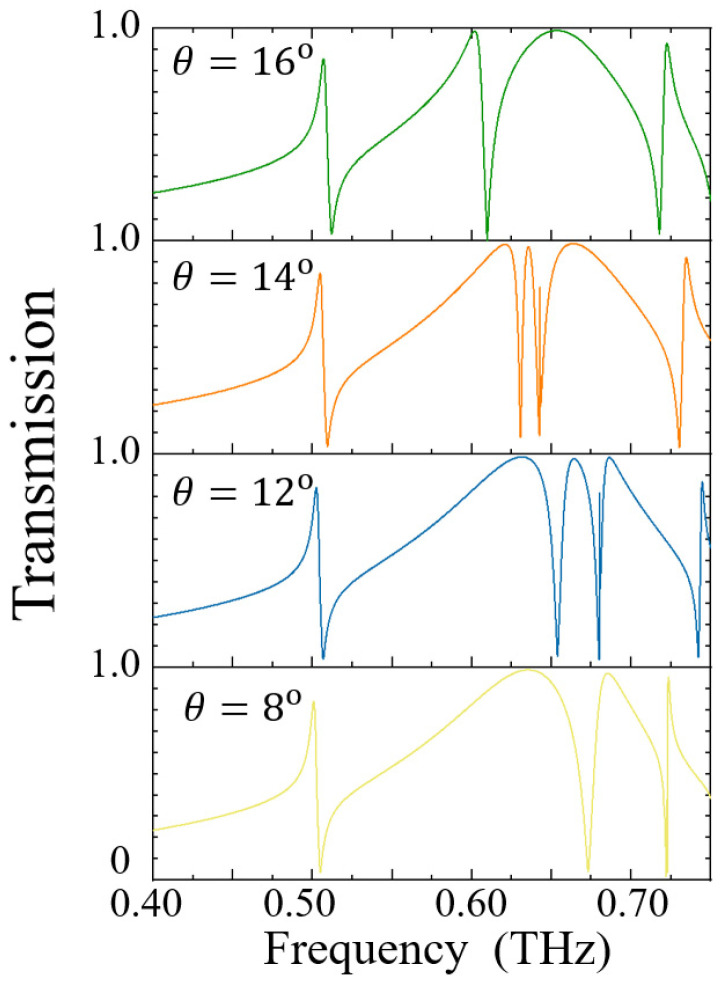
The transmission spectral characteristics of metal metasurfaces at different incident angles.

**Figure 5 sensors-25-04178-f005:**
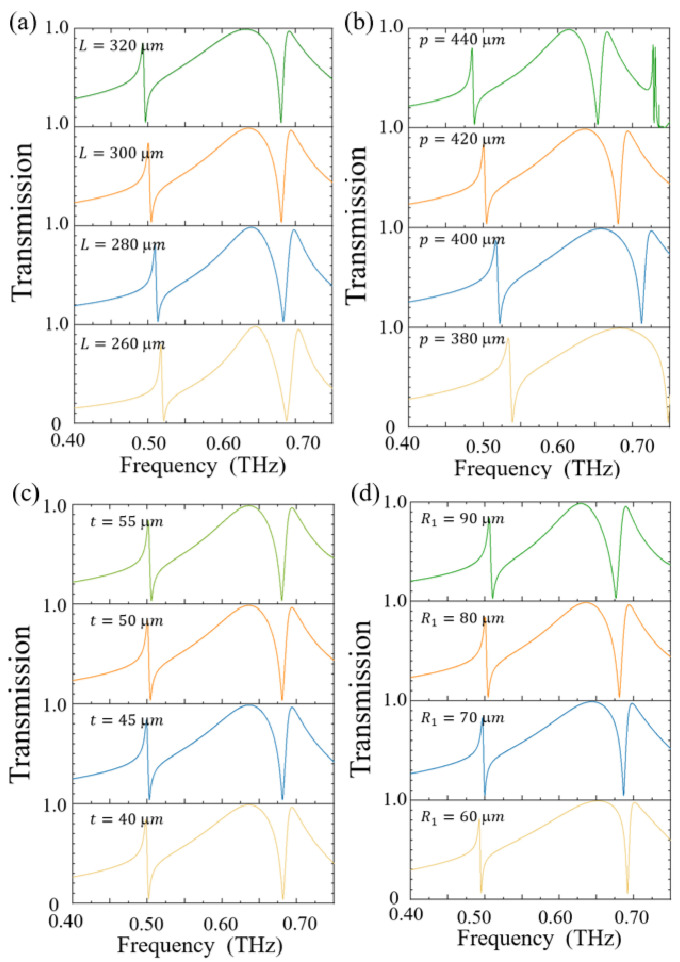
The influence of different structural parameters on the transmission spectrum of metasurface: (**a**) the influence of the side length L of the hollow rectangle; (**b**) the influence of the period P; (**c**) the influence of the metasurface thickness t; (**d**) the influence of the fillet R_1_.

**Figure 6 sensors-25-04178-f006:**
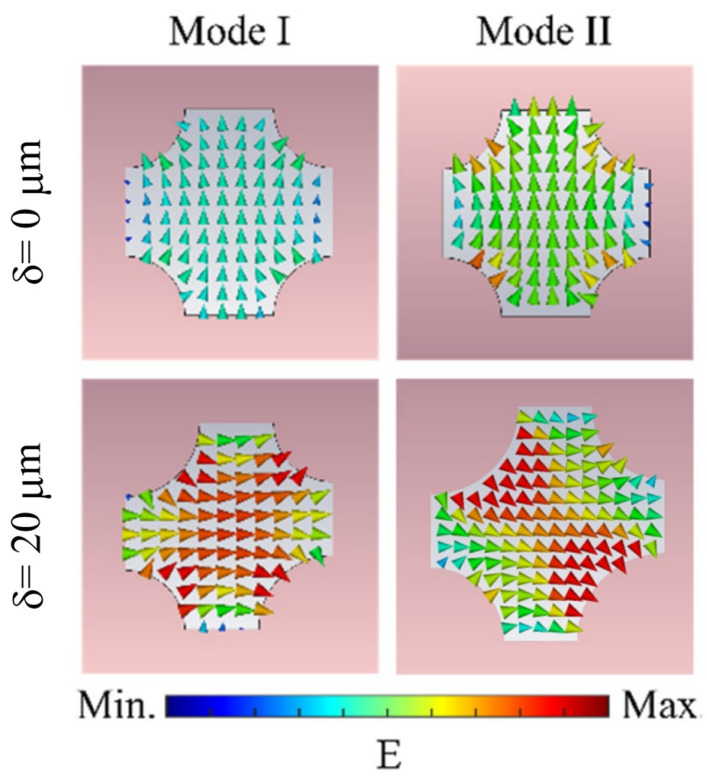
Electric field distribution of the metasurface in BIC and quasi-BIC states.

**Figure 7 sensors-25-04178-f007:**
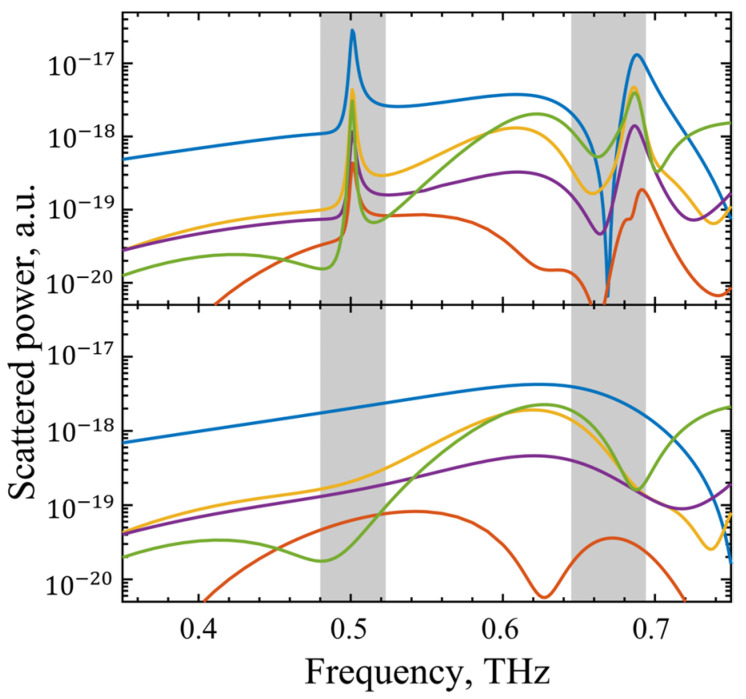
Scattering energy distribution of multipoles: (**above**) initial metamaterials; (**below**) symmetry-broken metamaterials. The five strongest terms, including electric dipole P, magnetic dipole M, toroidal dipole T, electric quadrupole $Q_{e}$, and magnetic quadrupole $Q_{m}$, are depicted.

**Figure 8 sensors-25-04178-f008:**
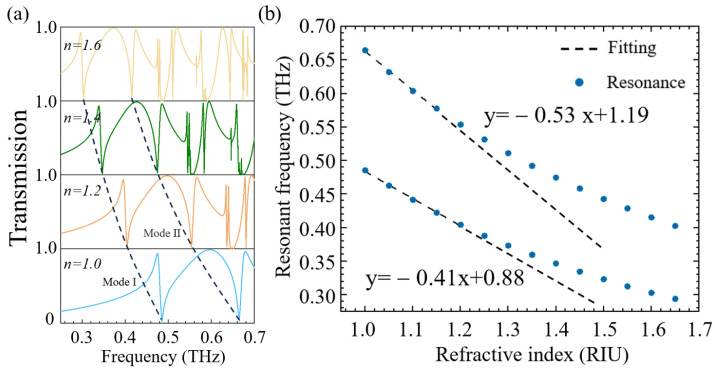
Transmission spectra at different refractive indices and their sensitivity: (**a**) transmission spectral lines at different refractive indices of the object to be measured; (**b**) the relationship between resonant frequency and refractive index.

**Figure 9 sensors-25-04178-f009:**
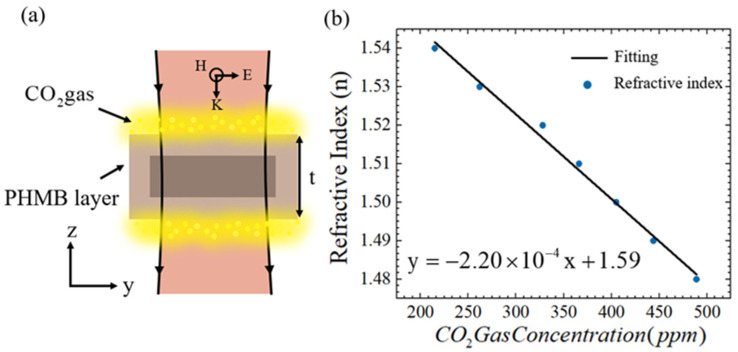
(**a**) Schematic diagram of CO_2_ gas sensing on the metasurface after deposition of the PHMB functional layer; (**b**) relationship between CO_2_ concentration and the refractive index of the PHMB functional layer [44].

**Figure 10 sensors-25-04178-f010:**
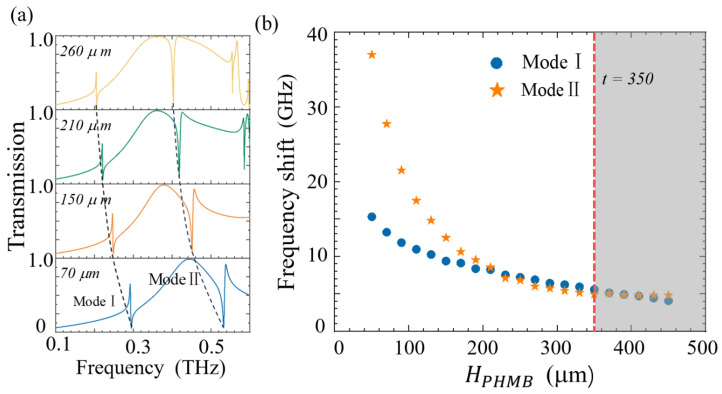
(**a**) Influence of the thickness of the deposited PHMB functional layer on the metasurface’s transmission spectrum; (**b**) the relationship between the thickness of the PHMB functional layer and the resonance offset.

**Figure 11 sensors-25-04178-f011:**
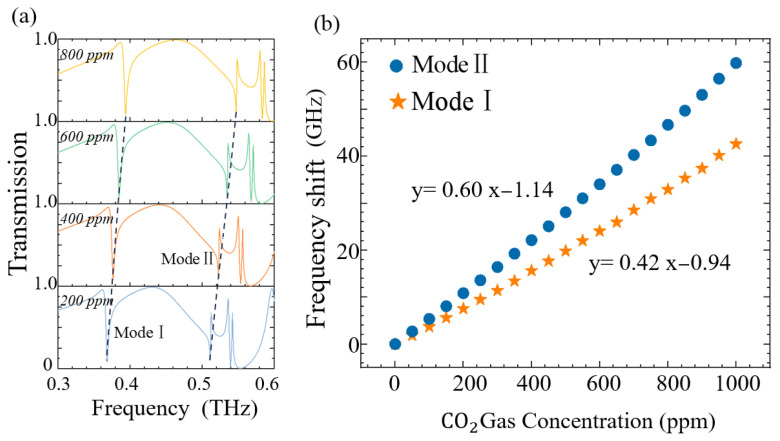
(**a**) When $H_{PHMB}$ = 350 μm, the influence of CO_2_ gas concentration on the metasurface’s transmission spectrum; (**b**) the relationship between CO_2_ gas concentration and resonance offset.

**Table 1 sensors-25-04178-t001:** Performance comparison of different sensors.

Sample	Material of Structured Layer	Frequency	Sensitivity (THz/RIU)	Sensitivity (Hz/RIU/F)	Reference
1	LaTiO_3_	2.170 THz	0.438	0.2018	[37]
2	Gold	2.020 THz	0.420	0.2079	[15]
3	Silicon	0.773 THz	0.231	0.2988	[38]
4	Graphene	4.200 THz	1.687	0.4017	[39]
5	Graphene	0.950 THz	0.770	0.8105	[40]
6	Tungsten	0.500 THz 0.670 THz	0.405 0.579	0.8092 0.8635	Our work

## Data Availability

The data involved in this study can be obtained by contacting the corresponding author and will not be publicly disclosed at this time.

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
