# Peer review of "Carbon Dioxide Gas Sensor Based on Terahertz Metasurface with Asymmetric Cross-Shaped Holes Empowered by Quasi-Bound States in the Continuum"

_sensors, 2025, doi:10.3390/s25134178_

Round 1

Reviewer 1 Report

Comments and Suggestions for Authors

The authors propose a quasi-bound state in the continuum (quasi-BIC) metasurface based on an asymmetric cross-shaped tungsten structure. FEM-simulations demonstrated that breaking the in-plane symmetry of the metasurface system enables the excitation of dual quasi-BIC resonances with very high quality factors.  The authors have explored the sensing applications of the structure by integrating into the metasurface a functional polymer layer comprising polyhexamethylene biguanide for the detection of carbon dioxide. Although an experimental validation is not given, the results are interesting. Since the ultrahigh Q factor of quasi-BICs can efficiently enhance the strength of light-matter interaction, the proposed design is interesting and can be useful for applications in terahertz biosensing and highly sensitive gas sensing.

However, before a final decision, a few points should be clarified to improve the paper:

1) The investigation considers only normal incidence. It should be interesting for the reader to see results on transmission in the case of oblique incidence and how the resonances are affected.

2) The authors state that FEM simulation method is used. Have you used a commercial software package or homemade code in the frequency domain? Which technique is utilized to obtain the optimized metasurface?

3) Please comment on the model you have used for the frequency-dependent complex permittivity of tungsten.

4) Could you please comment on the fabrication issues of your asymmetric metasurface, emphasizing the possible imperfections that deteriorate the very sensitive high-Q resonances?   

5) The caption of Figure 5 gives an inaccurate description and should be improved. In Figure 6, please write ‘scattered energy’ instead of ‘scattering energy’.

6) The authors should read the submitted manuscript and correct some minor English language syntactic errors.

Author Response

Dear reviewers

We appreciate for your work and constructive comments. The manuscript has been revised carefully according to the reviewers’ comments, please see the attached file. In our response, we list our point-by-point responses to the reviewers’ comments. The reviewers’ comments are in black, while our responses are in blue. In the revised manuscript submitted online, we labelled all changes in red.

Sincerely

Tian Ma

Reviewer 2 Report

Comments and Suggestions for Authors

Authors  propose a THz metallic metasurface with cross-shaped holes, which can excite dual quasi-BIC resonance modes with high quality factors by breaking symmetry. These resonances are driven by electric dipoles and electric quadrupoles, achieving refractive index sensitivities of 404.5 GHz/RIU and 578.6 GHz/RIU. Additionally, incorporating the functional material HMBB enables highly sensitive detection of CO2 gas concentrations.

I agree that the authors' simulations are reliable and of significant interest to readers of Sensor. However, I found a certain claim unclear and somewhat confusing. The authors emphasize that breaking symmetry enables the excitation of high-quality dual quasi-BIC resonance modes, but the proposed structure exhibits symmetry with respect to 45-degree directions. I suspect that the key point might be that different modes appear adjacent to each other for a specific polarization parallel to the x-axis and y-axis. In this regard, it seems similar to J. Phys. D: Appl. Phys. 55 (2022) 045302.
Therefore, I consider that the absence of polarization dependence in the x and y directions in Fig. 5 is arbitrary. The authors have to shows the spectra for  +/- 45-degree polarization directions in Figure 5.

Reviewer 3 Report

Comments and Suggestions for Authors

The article proposes a terahertz metasurface based on quasi-bound states in the continuum, and uses a finite element simulation software to calculate and analyze the electromagnetic characteristics of the metasurface. The unit structure is a cross-shaped hole etched on a tungsten metal plate with a period of 400 μm, formed by four fan-shaped notches. The symmetry is broken by changing R2 to excite the quasi-BIC response. Periodic boundary conditions and perfectly matched layers are adopted to reduce the boundary reflection. By fitting the Fano resonance model, the resonance frequency, linewidth are extracted and the Q factor is calculated, and Q is inversely proportional to the square of δ.

  1. How can the scheme proposed in the paper be actually processed? The whole paper only presents rationalized simulation results. For example, what is the method of changing R2? Once the processing is finalized, can this value no longer be changed?
  2. What is the basis for processing this shape? On what basis is this structure selected? What is the basis for setting the specific detail parameters of this structure? Why are such parameters set?
  3. What is the basis for the material selection? Is it related to the frequency in Figure 1 of the author? What are the advantages of tungsten metal for terahertz regulation? The paper discusses the spectral and resonance characteristics: when δ is 0 μm, the metasurface has C2 symmetry, supports the BIC state, and the linewidth of the transmission spectrum is zero; when δ is not 0, the C2 symmetry is broken, and a Fano-type resonance quasi-BIC state with a finite Q factor is generated, such as the excitation of double resonances at 0.50 THz and 0.68 THz. Increasing δ causes the resonance frequency to redshift and the linewidth to broaden, and the Q factor can be adjusted.
  4. How can δ be regulated and increased? In addition, the electric field and resonance mechanism are discussed: from the BIC state to the quasi-BIC state, the electric field distribution changes. Bode I and Bode II respectively discuss the multipole decomposition in the paper. The BIC state is mainly contributed by the electric dipole. In the quasi-BIC state, the electric dipole strength of Bode I is enhanced, and the electric quadrupole response of Bode II increases, verifying that the quasi-BIC resonance is mainly mediated by the electric field component of the terahertz wave, and the metasurface has polarization insensitivity.
  5. The polarization insensitivity is directly related to the designed structure and does not require special discussion. Finally, the author uses this metasurface as a refractive index sensor. When the environmental refractive index changes, the resonance frequency significantly redshifts. The refractive index sensitivities of Bode I and Bode II reach 404.5 GHz/RIU and 578.6 GHz/RIU respectively, and the quality factors are 33 and 32 respectively, which are superior to traditional terahertz sensors. After introducing the HMBB functional layer, it is used to detect carbon dioxide. HMBB has a selective adsorption of carbon dioxide, and its refractive index changes with the concentration of carbon dioxide, causing the resonance frequency of the metasurface to change accordingly. The optimal thickness of the HMBB layer is determined to be 350 nm. At this time, the detection sensitivities to carbon dioxide are 0.06 GHz/ppm (Bode I) and 0.04 GHz/ppm (Bode II), the detection limit is 33.33 ppm, and it shows a linear response within the concentration range of 0 - 1000 ppm, which is suitable for environmental monitoring.
  6. Here, the problems are more prominent. Firstly, has this device been actually processed? If it has been processed, the processing technology should be described in detail. If it has not been processed, the subsequent discussions and comparisons are meaningless. As for being superior to traditional terahertz sensors, that is, Table 1 mentioned by the author, the comparison of the literature is not comprehensive. It is recommended to discuss at the same frequency because it is actually difficult for an actual terahertz source to reach around 0.5 THz. Generally speaking, discussing its application in practice based on ideal simulation discussions and designs requires a change in this discussion mode. Otherwise, it is very easy to cause misunderstandings.

Reviewer 4 Report

Comments and Suggestions for Authors

The article report on transmission spectrum of an easy-to-fabricate metasurface that demonstrates sharp features in transmission due to occurrence of the quasi-BIS states. The paper presents detailed theoretical analysis of the electromagnetic response of the suggested metasurface including some qualitative considerations making it easy to grasp the main idea. Applications of the results are illustrated with CO2 sensing, based on the sensitivity of the resonant frequency to the refractive index of a coating with which the metasurface is covered.

While the main idea is clearly presented and is of importance to the Sensors readership there are several questions that have to be addressed:

  1. What is the suggested thickness of the metasurface?
  2. What is the role of the metasurface thickness? Will the results be different for 1 micron, 100 micron or 1 mm thick metasurfaces?
  3. It is next to impossible to arrange for a unform conformal coating of the metasurface with a polymeric film described in the reference 45. Authors should evaluate the effect of the film thickness non-uniformity and consider the effect of dielectric placed onto single surface of the structure.
  4. The metasurface is made of tungsten. What properties of tungsten are critical for the simulated results? What if it is replaced by copper or silver?
  5. The sensing effect is described via the resonant frequency shift. While more realistic experiments imply measurement of transmission. Authors should analyze the minimal concentration of the analyte based on a realistic NEP of a detector.

Reviewer 5 Report

Comments and Suggestions for Authors
  1. There are many problems in the introduction section. In terms of applications, although the applications of BIC in fields such as optics, acoustics, and electromagnetics, as well as in directions like high-sensitivity sensors and enhanced nonlinear optical effects, are mentioned, specific examples are lacking, making it difficult to intuitively understand its actual application scenarios and effects.
  2. The citation notations in the introduction are too brief. Only the numbers are listed, and the detailed information of the references, such as the authors, article titles, and journal names, is not provided, making it difficult for readers to trace and access the original literature. In addition, these citations are not closely connected to the subsequent research content, and neither can they fully demonstrate the supporting role for the research nor highlight the innovative points of this study.
  3. From the perspective of content logic, when elaborating on the metasurface, BIC, carbon dioxide monitoring, and other contents in the introduction, the transitions are rather abrupt, and the coherence is poor. It is difficult for readers to sort out a clear context of the research background. Meanwhile, there are spelling mistakes in the text. For example, "Mowever" should be corrected to "However", and "Heriodic" should be "Periodic". These mistakes affect the professionalism of the paper. Some expressions are vague. For instance, "such interactions" does not clearly indicate the specific referents. Moreover, there is a problem of repetitive and redundant expressions. For example, "The proposed quasi - BIC metasurface exhibits exceptional performance metrics, including a high Q - factor (> 10³)" and the previous "Theoretical simulations demonstrate that...dual quasi - BIC resonances with ultra - high quality factors (Q - factor> 10³)" repeat the description of the high Q - factor, making the text less concise and refined.
  4. There is a logical conflict in the setting of structural parameters in the paper. When describing the structure, it was initially set that "R1 = 80μm and R2 = 80μm". According to this setting, δ = R2 - R1 = 0. However, it is mentioned later that "when δ ≠ 0, the quasi - BIC response is excited", which is contradictory. The author should clarify the initial setting values of R1 and R2 and their change logic to ensure the consistency of the theoretical model.
  5. The paper proposes a terahertz metasurface based on an asymmetric cross - shaped hole for quasi - BIC resonance and sensing applications. However, in the introduction section, when compared with the related work of others, the innovative advantages of this study are not clearly highlighted. To enhance the attractiveness and value of the research, it is necessary to further clarify and elaborate on the innovative points of this study.
  6. In the process of exploring the optical properties and sensing mechanisms of the metasurface, the explanations of some key theories are insufficient. Taking the explanation of the excitation of the quasi - BIC resonance mode and the sensing principle as an example, the paper lacks in - depth theoretical analysis such as impedance matching in the sensor working principle. It is recommended to supplement such theoretical content, analyze the working mechanism of the metasurface from a more comprehensive and in - depth perspective, and enhance the integrity and persuasiveness of the theory.
  7. The overall language expression of the paper is relatively clear, but in some parts, the discrimination of similar concepts is insufficient. For example, when describing the characteristics, electric field distributions, and sensing principles of different quasi - BIC resonance modes (Bode I and Bode II), the expression methods are similar, making it difficult for readers to quickly and accurately distinguish the differences between them, thus affecting the understanding of the research content. It is recommended to optimize the relevant expressions and highlight the essential differences between different modes.
  8. When analyzing the influence of structural parameters (such as the asymmetry parameter δ) on the performance of the metasurface (such as the Q - factor, resonance frequency, and sensing sensitivity), the description in the paper is relatively superficial, mostly stating phenomena, and lacking in - depth logical deduction and analysis. This makes the scientificity and reliability of the research conclusions somewhat deficient. It is recommended to deeply analyze the internal relationships and action mechanisms among various parameters, supplement the logical deduction process, and improve the depth and credibility of the research.
  9. The paper cites few references from the past three years, failing to reflect the latest research trends and achievements in the field of terahertz metasurfaces in a timely manner. This field is developing rapidly with continuous emergence of new research results. Without the support of the latest literature, the elaboration of the research background and current situation will be insufficient, and the front - line nature and innovation of the research will also be affected. It is recommended that the author add high - quality references from the past three years, supplement the latest research progress, and make the research content more time - sensitive and comprehensive.
  10. There are problems in the editing and layout of the paper, which affect the standardization and professionalism of the paper. The symbol annotations in some of the figures and tables are non-standard and unclear, and there are situations where certain dimension annotations are missing or their positions are ambiguous in the structural schematic diagrams. The author needs to carefully check and revise the entire paper to ensure the format is standardized and unified.

Reviewer 6 Report

Comments and Suggestions for Authors

Authors presented a Polarization-Insensitive Terahertz Metasurface with Asymmetric Cross-Shaped Holes for Quasi-Bound States in the Continuum. This design is of some innovation. However, sample fabrication is necessary to verify the simulation result. Major revision is needed before accepted.

1. In the simulation part, the details of the simulation model were inadequately described. For example, the setting of material parameters and the selection of boundary conditions in the model have a significant impact on the simulation results. Authors should describe these parameters in detail.

2. Although the advantages of structural design in fabrication were emphasized, there is a lack of detailed description and feasibility assessment of the actual fabrication process. Authors should add the specific process steps, key process parameters, and possible technical challenges in the fabrication process of this metasurface.

3. Regarding the CO₂ sensing, the sensing performance of CO₂ was only based on literature research and simulations. Experimental research should be added to verify the sensitivity of this work.

Round 2

Reviewer 1 Report

Comments and Suggestions for Authors

The authors have followed my suggestions carefully and responded adequately to my comments.

The required figures and explanations/corrections have been added.

The paper has been significantly improved and can be published in 'Sensors' in its current form.  

Author Response

Thank you very much for your positive assessment and acceptance of our manuscript. We sincerely appreciate your time and valuable insights throughout the review process.

Reviewer 2 Report

Comments and Suggestions for Authors

In the previous round of review, I generally agreed with the authors' works. However, I pointed out that the designed metasurface exhibits anisotropy and strong polarization dependence when the optical axis is tilted by 45°. The authors agreed with this point and revised their manuscript. After carefully reading, I recommend that the authors should revise their manuscript carefully before publication.

For example, the abstract remains unchanged, yet its first sentence still presents the issue as follows. Similarly, the conclusion contains a statement that has not been revised in response to the previous reviewers' comments. 

Additionally, the modified sections highlighted in red are particularly difficult to read. For example, the following phrases require attention:

- "In practical" → Should be corrected to "In practice".

- "the condition γ > instrument resolution" → Consider rephrasing for clarity, such as "ensuring that the resonance linewidth (γ) exceeds the instrument resolution."

- "the optical properties of the proposed metasurface were theoretically calculated via numerical simulations conducted by the CST Microwave Studio 2021 software." → The phrase "theoretically calculated via numerical simulations" is somewhat redundant. A more concise version could be "The optical properties of the proposed metasurface were analyzed using numerical simulations performed with CST Microwave Studio 2021."

- "using direct laser cutting technology" → Consider simplifying to "using laser cutting technology" or "via direct laser cutting."

- "As shown in Figure 10a, increasing 𝐻𝑃𝐻𝑀𝐵 from 0 to 350 nm" → Should "nm" be corrected to "μm"?

- "(a) When HPHMB = 350 nm," in Figure caption 11 → Should "HPHMB" be formatted as "H_{PHMB}"? Also, should "nm" be corrected to "μm"?

Author Response

Thank you very much for your positive assessment and acceptance of our manuscript. We carefully proofread our manuscript and revised typos and mistakes in language.

Reviewer 3 Report

Comments and Suggestions for Authors

Based on its current content, the paper is deemed acceptable for publication in its present form.

Author Response

(The authors gave the same response as above.)

Reviewer 4 Report

Comments and Suggestions for Authors

My comments were properly accounted for 

Author Response

(The authors gave the same response as above.)

Reviewer 5 Report

Comments and Suggestions for Authors

The revised manuscript has been carefully examined. All the problems have been properly solved. This article has a rigorous structure and smooth logic. It is now ready to move on to the next stage.

Author Response

(The authors gave the same response as above.)

Reviewer 6 Report

Comments and Suggestions for Authors

Authors response to the concerning quesitons. Although experiment research has not been added in this manuscript, the detail simulation and principle has been well discussed. Authors could continue the  experiment research in further work. 

The manusript could be accepted.

Author Response

(The authors gave the same response as above.)
